# Resilience Interventions Conducted in Western and Eastern Countries—A Systematic Review

**DOI:** 10.3390/ijerph19116913

**Published:** 2022-06-05

**Authors:** Manpreet Blessin, Sophie Lehmann, Angela M. Kunzler, Rolf van Dick, Klaus Lieb

**Affiliations:** 1Leibniz Institute for Resilience Research (LIR), 55122 Mainz, Germany; manpreet.blessin@lir-mainz.de (M.B.); sophie.lehmann@lir-mainz.de (S.L.); angela.kunzler@lir-mainz.de (A.M.K.); 2Department of Social Psychology, Goethe University, 60323 Frankfurt, Germany; van.dick@psych.uni-frankfurt.de; 3Department of Psychiatry and Psychotherapy, University Medical Center Mainz, 55131 Mainz, Germany

**Keywords:** resilience, intervention, cultural psychology, mental health, anxiety, depression, quality of life, perceived stress, social support

## Abstract

Previous research has demonstrated the efficacy of psychological interventions to foster resilience. However, little is known about whether the cultural context in which resilience interventions are implemented affects their efficacy on mental health. Studies performed in Western (*k* = 175) and Eastern countries (*k* = 46) regarding different aspects of interventions (setting, mode of delivery, target population, underlying theoretical approach, duration, control group design) and their efficacy on resilience, anxiety, depressive symptoms, quality of life, perceived stress, and social support were compared. Interventions in Eastern countries were longer in duration and tended to be more often conducted in group settings with a focus on family caregivers. We found evidence for larger effect sizes of resilience interventions in Eastern countries for improving resilience (standardized mean difference [*SMD*] = 0.48, 95% confidence interval [*CI*] 0.28 to 0.67; *p* < 0.0001; 43 studies; 6248 participants; *I*^2^ = 97.4%). Intercultural differences should receive more attention in resilience intervention research. Future studies could directly compare interventions in different cultural contexts to explain possible underlying causes for differences in their efficacy on mental health outcomes.

## 1. Introduction

Resilience describes the maintenance or fast recovery of mental health during or after substantial adversities [1]. The latter might include a cumulation of micro-stressors (e.g., daily hassles such as traffic jams) or major life events (i.e., macro-stressors like the loss of a loved one) [2,3,4,5]. Resilience is a lifelong, ongoing process that does not necessarily lead to a person encountering fewer stressors in their life, but can lead to more effective coping with stressors and more adaptive responses [3]. The most recent approach in social and health science conceptualizes resilience as a positive outcome, i.e., maintaining or quick regaining mental health during or after adversities. In this concept, adaptation processes that finally lead to resilience outcomes are called resilience processes. Those resilience processes are facilitated by resilience factors. Resilience factors may be psychosocial factors (e.g., active coping, self-efficacy, optimism, social support and hope), but also (epi)genetic, neurobiological, immunological, or other biological factors [6], which are associated with each other and may interact with each other.

Psychology has aimed at fostering resilience with different interventions, that are as diverse as the stressful situations that people face [4,7,8,9,10]. Resilience interventions have to take place before an anticipated stressor, during or after stressors, as resilience can only be seen in relation to a stressor [11,12]. In previous intervention research, resilience-training programs usually focused on strengthening one or several resilience factors to foster the outcome of resilience, using different approaches such as mindfulness-based therapy, problem-solving training, or elements of positive psychology [11,13]. Studies to evaluate resilience interventions often do not focus on resilience as an outcome but instead measure related outcomes such as quality of life or measures of stress and negative affect (e.g., anxiety, depression, perceived stress) [11]. Previous research on resilience interventions already demonstrated the efficacy on such mental health outcomes in different populations. Potential moderators for the efficacy of resilience-training programs have been investigated, showing that different aspects of interventions (e.g., group versus individual sessions, duration of the intervention, or format of delivery) may affect their efficacy [13,14].

One important gap in previous resilience intervention research that the present review intends to address is the lack of a cultural perspective [15,16].

### Purpose of This Review

Henrich et al. [17] already described that psychological research is often explored in a Western, educated, industrial, rich, and developed context (the so-called WEIRD problem). Given that large parts of the global population are not WEIRD, the focus on WEIRD contexts is likely to cause bias [17,18,19]. As Ungar mentioned in his work, resilience research is not only conducted exclusively for Western populations but also does not consider resilience definitions from other cultures or even what is considered healthy functioning in different cultures [16]. For example, mental health can not only appear as a mental construct, but also as a physical construct when it comes to somatization of mental health problems [20,21,22], which may differ across cultures [23,24,25]. Regarding the WEIRD problem, resilience research should also consider that different cultural contexts can lead to different stressors and may even require different resilience factors that need to be acknowledged and fostered in other, better fitting, ways. Therefore, this review aimed to address this research gap and to draw attention to the WEIRD problem by contrasting resilience interventions for adult populations in Western and Eastern countries.

This review does not aim to qualitatively describe resilience intervention differences in studies from Western versus Eastern countries but gives a first, broad overview on how resilience interventions are implemented in different countries which can help to explore this research gap further. With this knowledge, future resilience interventions may address the cultural aspect of different stressors, resilience definitions, and various ways of describing mental health symptoms in a more specific way.

## 2. Materials and Methods

In this review we compared different aspects of resilience interventions between training programs conducted in Western versus Eastern countries, specifically study settings, mode of delivery, target populations, underlying theoretical approaches, durations of training, and the study design by regarding the control group designs that were used. We also contrasted their efficacy for a range of mental health outcomes. More specifically, the effects of resilience-training programs conducted in Western and Eastern countries on the outcomes of resilience, anxiety, depressive symptoms, quality of life, perceived stress, and social support were examined in this review. We assessed both resilience and resilience-related outcomes, as resilience is no unified construct in the current and past research field, and we did not want to exclude studies just because they had a different wording for a similar construct in different countries and time periods.

This review was conducted based on a dataset which has been identified and analyzed for a systematic Cochrane review on psychological interventions to foster resilience in adults. The Cochrane review has been planned and performed to systematically identify resilience-training programs in various populations. The overall dataset consists of 221 randomized controlled trials (RCT), including cluster-RCTs, published between January 1990 and June 2019, which had been identified by searches performed in October 2016 and June 2019. To date, the evidence identified for resilience-training programs in healthcare workers (44 RCTs) [12] and healthcare students (30 RCTs) [13] has been published in two Cochrane publications.

Considering grey literature and ongoing studies (e.g., from trial registrations), the dataset of 221 RCTs, which was used for this review, was retrieved from the following 17 electronic sources, using a search filter for RCTs and limited to the period from 1990 onwards: Cochrane Central Register of Controlled Trials (CENTRAL), MEDLINE Ovid, Embase Ovid, PsycINFO Ovid, CINAHL EBSCOhost (Cumulative Index to Nursing and Allied Health Literature), PSYNDEX EBSCOhost, Web of Science Core Collection, International Bibliography of the Social Sciences ProQuest (IBSS), Applied Social Sciences Index & Abstracts ProQuest (ASSIA), ProQuest Dissertations & Theses (PQDT), Cochrane Database of Systematic Reviews (CDSR), Database of Abstracts of Reviews of Effects (DARE; only in October 2016), Epistemonikos, ERIC EBSCOhost, and the trial registers World Health Organization International Clinical Trials Registry Platform (WHO ICTRP), ClinicalTrials.gov and ISTRCN registry.

The search strategy for the two searches in MEDLINE as an example is presented in the Appendix A, Appendix A.

### 2.1. Inclusion Criteria

The studies included in the present review had to fulfill the following criteria: Participants were required to be adults aged 18 years or older who were exposed to a stressor in the past, were currently facing a stressor, or were anticipated to be exposed to substantial adversities in the future. We included psychological resilience interventions, that is, interventions focusing on fostering resilience or related concepts (e.g., hardiness or posttraumatic growth) [12,26]. Outcomes for this review included resilience and different outcomes of mental health: anxiety, depressive symptoms, quality of life, and perceived stress as well as perceived social support.

The search was not restricted to any publication language. Studies published in Chinese or Iranian were translated in full length by native or experienced foreign speakers of these languages.

### 2.2. Study Selection

The results of the study selection process based on the systematic Cochrane review are shown in Figure 1 using a PRISMA flow diagram. After deduplication, 21,237 references were screened by title and abstract by two reviewers working independently. Subsequently, 2163 full-text articles were assessed for eligibility in duplicate by the same reviewers. Finally, 221 articles met the eligibility criteria and were included in this review. Data of included RCTs were extracted by Kunzler and colleagues [26], using an electronic data extraction sheet based on the Cochrane guidelines [27] and by two reviewers working independently. Any disagreements in study selection or data extraction were resolved by discussion or by consulting a third reviewer.

### 2.3. Data Analysis

The included studies were synthesized in a narrative and tabular form. All studies were clustered into Western or Eastern countries depending on the country in which they were conducted. This review includes a qualitative synthesis by summarizing the characteristics of the included studies (i.e., setting, mode of delivery, target population, theoretical approach, duration, control group design). In addition, we performed pairwise meta-analyses for all mental health outcomes considered in this review.

Data were analyzed using Microsoft Excel [28], RevMan Web [29] and R [30] with the following packages: *readxl* [31], *xlsx* [32], *dplyr* [33], *meta* [34], *lsr* [35], and *metafor* [36].

To compare resilience interventions in Western versus Eastern groups with respect to intervention characteristics such as setting, mode of delivery, target population, underlying theoretical approach, duration and control group design, Chi-Square (*χ*^2^) and Cramer’s *V* were calculated via the *lsr* package in R [30,35].

Pairwise meta-analyses (i.e., comparing resilience interventions and control groups), specifically subgroup analyses, were performed to identify pooled intervention effects of resilience interventions in Western versus Eastern countries and to compare their efficacy on various mental health outcomes. We calculated both the pooled effect size across all available studies (i.e., Western and Eastern countries) for a specific outcome and performed a subgroup test contrasting the studies from both cultural contexts. We applied random-effects modeling to pool the effect sizes using the R package *metafor* [36] to account for between-study heterogeneity regarding outcome measures, as recommended in most clinical psychology and health sciences [37]. Standardized mean differences (*SMDs*) and their 95% confidence intervals (*CI*) were calculated using the generic inverse-variance approach from the meta and package in R [30,34,36]. All calculated *SMDs* refer to the respective outcome, a higher *SMD* indicates higher efficacy of the respective intervention on the mentioned outcome. The test statistics and *CIs* were adjusted with the Hartung-Knapp-Sidik-Jonkman method [38]. *I*^2^ was calculated to identify and measure heterogeneity between studies with the meta package in R. The Cochrane Handbook for Systematic Reviews of Interventions was used for the interpretation of *I*^2^: low (0–40%), moderate (30–60%), substantial (50–90%), and considerable heterogeneity (75–100%) [27,30,34]. Forest plots were generated using the meta package in R [30,34].

### 2.4. Risk of Bias

The quality assessment was performed with the Cochrane Risk-of-Bias Assessment Tool for Randomized Control Trials. The risk of bias graph and summary table was generated with RevMan Web [29].

We assessed random sequence generation (selection bias), allocation concealment (selection bias), blinding of participants and personnel (performance bias), blinding of outcome assessors (detection bias), incomplete outcome data (attrition bias), and selective outcome reporting (reporting bias). The included studies were rated for each assessment with a high, unclear, or low risk of bias.

## 3. Results

### 3.1. Characteristics of Included Studies

The 221 studies reviewed were conducted in 28 countries, including 174 studies from 18 Western countries (USA, Australia, Canada, The Netherlands, Germany, the United Kingdom, Spain, Israel, Denmark, Italy, Poland, Portugal, Sweden, Belgium, Cyprus, Finland, Hungary, New Zealand) and 47 studies from ten Eastern countries (Iran, China, India, Thailand, Dominican Republic, Sierra Leone, Singapore, South Korea, Sri Lanka, Taiwan); see Table 1.

Across all included studies, the intervention settings varied between group setting, individual setting, and a combination of both. The interventions were delivered using different formats: face to face, online, telephone, laboratory, smartphone, bibliotherapy, audio, and a combination of different formats.

The population groups targeted by the interventions were heterogenous and clustered into the following categories: employees in organizations of different branches, patients (physical health conditions), students—various fields, military/police, general population (e.g., volunteers), family caregivers, patients (mental health conditions), employees–teachers.

The included studies used different theoretical approaches, including mindfulness-based therapy [204], cognitive behavioral therapy (CBT) [161], attention & interpretation therapy (AIT) [127], stress inoculation training [258], problem-solving training [51], acceptance and commitment therapy (ACT) [259], positive psychology approaches [260], cognitive bias modification [50], and multimodal resilience trainings based on several theoretical approaches.

Training length of interventions could be clustered into high intensity (i.e., more than 12 h or 12 sessions), moderate intensity (i.e., 5 to 12 h or 3 to 12 sessions), and low intensity (i.e., less than 5 h total or 3 sessions).

The included RCTs used the following control group designs: waitlist control, attention control, treatment as usual (TAU), active control, no intervention, and a combination of different control group designs (Table 2).

Furthermore, the Appendix A provide an overview of all scales that were used to measure the outcomes of resilience, anxiety, depressive symptoms, quality of life, perceived stress, and social support.

Of all 221 included studies, 43 studies measured resilience as an outcome. In these studies, resilience was defined as a state or process most of the times (*n* = 41, assessed for example with CD-RISC or BRS), whereas only two studies defined resilience as a trait (assessed with the Cognitive Hardiness Scale by Nowack 1989 [263]). In these studies, the intervention mostly took place during a stressor (*n* = 39), and partly before an anticipated stressor (*n* = 1), after a stressor (*n* = 2), or unspecified (*n* = 2). Some stressors were normative (*n* = 13, e.g., workplace related stress, age-associated loss of resources or academic stress), and some were non-normative (*n* = 30, e.g., sudden severe illnesses, nature disasters or homelessness). The sample size of these 43 studies ranged from *n* = 22 to *n* = 918 with a mean sample size of *n* = 110.63 (Appendix A). The intervention programs were all RCTs, including *n* = 11 pilot studies.

### 3.2. Risk of Bias

Regarding risk of bias of the 221 included studies, the main limitations (≥20% high risk) were found in the following domains: blinding of participants and personnel for subjective (self-reported) outcomes (performance bias, 203 studies), incomplete outcome data (attrition bias, 125 studies), and selective outcome reporting (reporting bias, 70 studies), see Figure 2 and the Appendix A.

### 3.3. Differences in the Implementation of Interventions between Western and Eastern Populations

A statistically significant difference between interventions that were conducted in Western versus Eastern countries was found when looking at the target population (*χ*^2^(7) = 46.36, *p* < 0.001), with a moderate Cramer’s *V* of 0.46) [264]. While both Western and Eastern studies examined patients with physical health conditions, studies from Western countries focused on employees, whereas those from Eastern countries investigated family caregivers and the general population.

Concerning the duration of the interventions, we identified a statistically significant difference between Western and Eastern countries (*χ*^2^(3) = 15.07, *p* = 0.002; low Cramer’s *V* of 0.26). The distribution of durations between Western and Eastern interventions shows that there are hardly any low-intensity interventions conducted in Eastern countries.

There was a statistically significant difference between intervention settings used in Western and Eastern countries (*χ*^2^(3) = 18.8, *p* < 0.001, moderate Cramer’s *V* of 0.29) [264], showing that almost exclusively group interventions were conducted in Eastern countries.

There also was a statistically significant difference with regards to the study design aspect between the comparators used in Western and Eastern countries (*χ*^2^(6) = 20.81, *p* = 0.002, moderate Cramer’s *V* 0.31), showing a tendency of Eastern countries to use no intervention control groups more often than studies from Western countries do.

No statistically significant difference regarding the theoretical approach of interventions was found between Western and Eastern countries (*χ*^2^(9) = 16.75, *p* = 0.05; with a moderate Cramer’s *V* of 0.28). There was also no statistically significant difference between Western and Eastern countries with respect to the mode of delivery of the interventions (*χ*^2^(8) = 13.84, *p* = 0.086, moderate Cramer’s *V* of 0.25).

### 3.4. Differences between Western and Eastern Countries in Effect Sizes on Mental Health Outcomes

#### 3.4.1. Resilience

In total, 43 included studies measured resilience, 33 studies from Western and ten from Eastern countries, which could be included in the subgroup analysis. The pooled effect estimate showed evidence for an overall moderate positive effect (i.e., across all 43 studies from Western and Eastern countries) of interventions on resilience (*SMD* = 0.48, 95% *CI* 0.28 to 0.67; *p* < 0.0001; 4797 participants; *I*^2^ = 97.4%), Figure 3.

As shown in Figure 3 and Table 3, resilience interventions showed larger effect sizes on resilience in Eastern compared to Western countries, with the test for subgroup differences providing evidence for a significant difference (*Q*(1) = 5.61, *p* = 0.018).

#### 3.4.2. Anxiety

The meta-analysis for the outcome of anxiety included 29 studies. The pooled effect estimate provided evidence for an overall small effect of interventions on anxiety (*SMD* = −0.37, 95% *CI* −0.57 to −0.18; *p* = 0.0005; 2436 participants; *I*^2^ = 93.8%), forest plot in the Appendix A. The test for subgroup differences between 24 studies from Western countries and five studies from Eastern countries could not be conducted due to the small number of studies per subgroup (Table 3) [265].

#### 3.4.3. Depressive Symptoms

In the meta-analysis for depressive symptoms, 16 studies were included. The pooled effect estimate showed evidence for an overall moderate effect in favor of resilience-training programs on depressive symptoms (*SMD* = −0.46, 95% *CI* −0.62 to −0.29; *p* < 0.0001; 1706 participants; *I*^2^ = 90.0%), forest plot in the Appendix A. Based on the small total number of 16 studies assessing depressive symptoms, 14 studies conducted in Western and only two studies in Eastern countries, the number of studies per subgroup was too small to conduct any subgroup analysis (Table 3) [265].

#### 3.4.4. Quality of Life

Thirty-eight studies could be included in the meta-analysis for quality of life. The pooled effect estimate showed evidence for an overall small effect in favor of resilience interventions on quality of life (*SMD* = 0.29, 95% *CI* 0.15 to 0.43; *p* < 0.0001; 5745 participants; *I*^2^ = 98.1%), forest plot in the Appendix A. The test for subgroup differences, with 33 studies assigned to Western countries and five studies to Eastern countries could not be conducted due to the small number of studies per subgroup (Table 3) [265].

#### 3.4.5. Perceived Stress

In the meta-analysis regarding the outcome of perceived stress, we could include 37 studies. The pooled effect estimate provided evidence for an overall moderate effect in favor of resilience interventions on perceived stress (*SMD* = −0.42, 95% *CI* −0.62 to −0.22; *p* < 0.0001; 2918 participants; *I*^2^ = 92.0%), forest plot in the Appendix A. No subgroup analyses could be performed between the 35 studies from Western countries and the two studies from Eastern countries due to the small number of studies per subgroup (Table 3) [265].

#### 3.4.6. Social Support

Overall, four studies could be included in the meta-analysis for social support. There was no evidence for any overall effect of resilience interventions on social support (*SMD* = 0.05, 95% *CI* −0.44 to 0.54; *p* = 0.7644; 1589 participants; *I*^2^ = 99.2%), forest plot in the Appendix A. Due to the small number of studies measuring social support, two studies in Western countries and two studies in Eastern countries, no subgroup analysis could be conducted (Table 3) [265].

## 4. Discussion

### 4.1. Principal Findings

This review aimed to explore potential differences in the efficacy of resilience interventions between Western and Eastern countries. Based on descriptive characteristics of the 221 included studies, resilience interventions in Western countries were mostly conducted for patients with physical health conditions and for employees. In Eastern countries, the most common target populations of resilience interventions included patients with physical health conditions, family caregivers, and the general population. While Western countries showed a broad range of different training durations, that is, from low to high intensity, Eastern countries tended to conduct almost only moderate- and high-intensity resilience interventions. Another difference could be found regarding the setting of resilience interventions: both, Western and Eastern countries showed a tendency to prefer group setting designs. However, considering the relation of the number of studies, this preference is even more pronounced in Eastern countries. This could be explained by one of Hofstede’s dimensions that describes how Western countries tend to be more individualistic and Eastern countries to be more collectivistic [266].

Additionally, the results showed significant differences between the control groups used in studies from Western and Eastern countries, demonstrating that intervention researchers in Eastern countries tended to use no intervention control groups more often, which needs to be considered with caution as the results could be distorted by this.

No differences were found concerning the theoretical approach of interventions. This highlights the WEIRD bias in resilience research and shows that Eastern countries also tend to implement interventions that are based on Western theoretical approaches of interventions. No significant differences were found in the mode of delivery used since both Western and Eastern countries tend to use face to face formats or a combination of different modes.

In addition to the above-mentioned descriptive differences in the implementation of resilience interventions and their evaluation, we identified evidence for larger effect sizes in favor of Eastern countries for the outcome resilience. No significant differences regarding the efficacy of resilience-training programs were found for other mental health outcomes or social support.

That this review does not demonstrate highly variable outcomes of resilience interventions in terms of whether they take place in Western or Eastern countries may indicate a cultural sensitivity for most interventions. They seem to fit in their cultural context, which could be a hopeful sign for planning future interventions more often especially in Eastern countries.

### 4.2. Comparison to the Literature

In recent years, the high number of systematic reviews and meta-analyses on the topics of resilience and resilience interventions demonstrated the high interest in this research field and identified aspects that are still open to be researched and discussed further [11,12,13,26,267,268,269]. Addressing in more detail whether resilience should be understood as a process or an outcome [268,270], for example, is of high importance, and this conceptualization also differed in the RCTs that were included in this review.

Another topic that seems to arise with this aspect is differences in the concept of resilience and therefore also resilience interventions in different countries and cultures giving the reader an overview of aspects that should be acknowledged or further researched in the future. As far as we know, none of the thus far conducted systematic reviews and meta-analyses in the resilience research field focused on differences between interventions that were conducted in different countries or cultures, although resilience across different cultures has already been discussed as an important aspect to be addressed [16,271].

### 4.3. Limitations

This review is subject to different limitations. As indicated by the risk of bias assessment, several of the included studies were at high risk of bias in various domains, which may distort the reported effects. Another limitation is the large between-study heterogeneity of the included studies with respect to, for example, the quality of the studies, the assessment scales for outcome measurement or the comparators used. This heterogeneity between studies could in part have resulted in the considerable statistical heterogeneity and variation of treatment effects between studies, as shown by large values of *I*^2^. Another point to consider is the difference in the number of studies conducted in Western and Eastern countries, which is reflected in the subgroup analysis for the outcome resilience. Moreover, due to the small number of subgroups in each outcome, we could not test for statistical differences in outcomes regarding study designs in Western or Eastern countries.

For this review, we only focused on post-intervention assessments since no sufficient data were available to conduct further analyses for follow-up time points. Furthermore, not all pre-planned analyses could be conducted, due to a lack of studies for specific subgroup analyses.

As suggested by Copas et al. [272], the reviewed studies include primary as well as secondary outcomes. We did not take into account in our analysis if the outcomes were originally assessed as primary or as secondary outcomes. Future research might compare if the efficacy of resilience and other outcomes differ when assessed as primary or as secondary outcomes in reviewed studies.

Resilience interventions that were conducted in Eastern countries were considerably influenced by Western research as the theoretical approaches showed, which could also have biased the results of this review. Although we are aware of this circumstance, we could not handle the data in a more culturally sensitive way due to the original focus of the studies, which did not intend to provide a cultural overview. Therefore, we had to work with the country label of each study and could not describe this aspect of the target populations further.

In the face of the COVID-19 pandemic and its related stressors (e.g., measures of containment), it can be anticipated that interventions to foster mental health and to prevent mental illness—including resilience interventions—will be needed even more in different population groups, with various training programs already currently being conducted and evaluated [273,274]. For this paper, these studies were not considered since most studies for that specific stressor are still ongoing. Nevertheless, future research on cultural differences should also consider interventions that were conducted in the aftermath of the disease outbreak (i.e., after 2020) with an updated literature search.

Furthermore, measurements of somatization should be included in future research to assess resilience in a more culturally sensitive way, especially in countries where mental health and mental health treatment are stigmatized. We emphasize the relevance for future research to address the WEIRD problem in resilience interventions.

### 4.4. Implications

This review revealed several research gaps that need to be addressed before resilience interventions can be satisfactorily compared across cultural contexts. This would require, for example, that individuals participating in a resilience intervention be interviewed by researches regarding various cultural aspects that they might—or might not—hold in [269]. It is not sufficient to infer the cultural values of the subjects from the country in which the resilience intervention took place, or from the cultural values of the researcher who conducted the resilience intervention.

There are two different aspects that need to be considered and hold different values in themselves (1) the efficacy of resilience interventions in different countries and (2) differences in resilience interventions in different countries. To explicitly show differences in the efficacy of resilience interventions in different countries it would be recommended to conduct a resilience intervention in different countries that delivers the same contents, has the same study settings and similar target populations to reduce heterogeneity between the resilience interventions. Results that show different efficacies of the same intervention could then help to highlight cultural differences and hint to study settings or even resilience factors that may be more efficient or relevant in some cultures than in others. At least as valuable, if not more so, would be to look at different resilience interventions not of Western origin conducted in different countries. A more qualitative approach could help to gather non-westernized intervention approaches or even resilience factors that might not have been considered by Western research. This could not only help to construct resilience interventions that could be more suitable for different populations but could also strengthen Western resilience interventions by incorporating new aspects that were not previously considered.

### 4.5. Conclusions

Different countries hold a great deal of potential in that other could learn from them and that could also help to understand psychological concepts as a whole. Our world is becoming more intercultural each day and as part of the sphere of Western countries and the accompanying WEIRD bias, we should rethink the way we consider different countries in psychological research. With respect to migration and refugee crises, it should be particularly important to develop resilience interventions that are tailored to the specific populations they are ultimately intended to reach. Culturally sensitive approaches could help to conduct resilience interventions that are more effective for specific populations and/or could help to extend our knowledge about effective resilience interventions.

## Figures and Tables

**Figure 1 ijerph-19-06913-f001:**
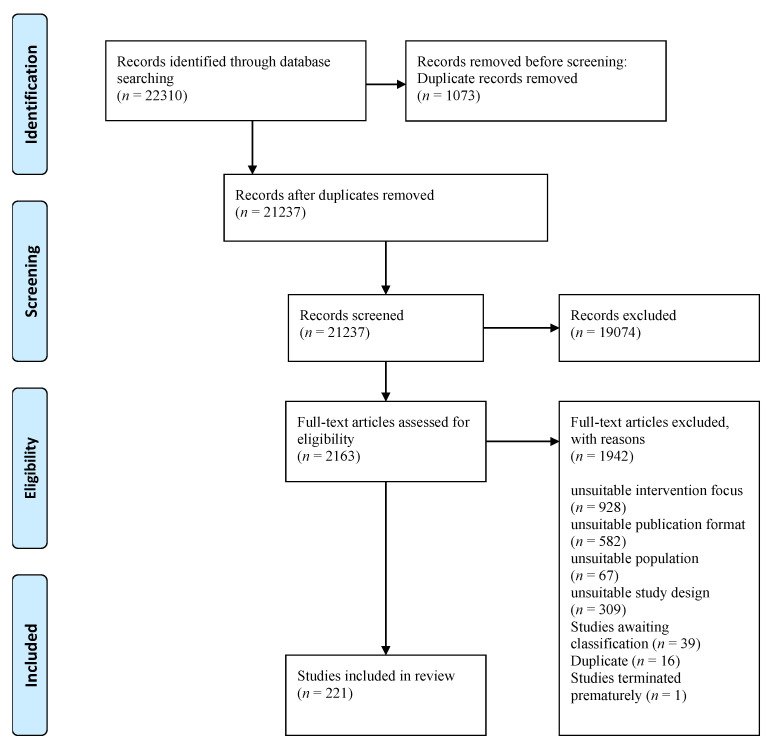
PRISMA flow diagram.

**Figure 2 ijerph-19-06913-f002:**
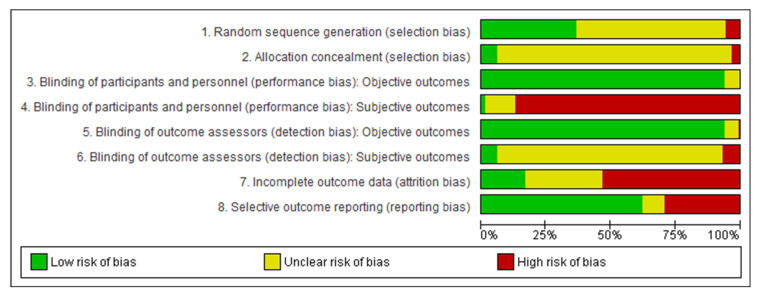
Risk of Bias Graph.

**Figure 3 ijerph-19-06913-f003:**
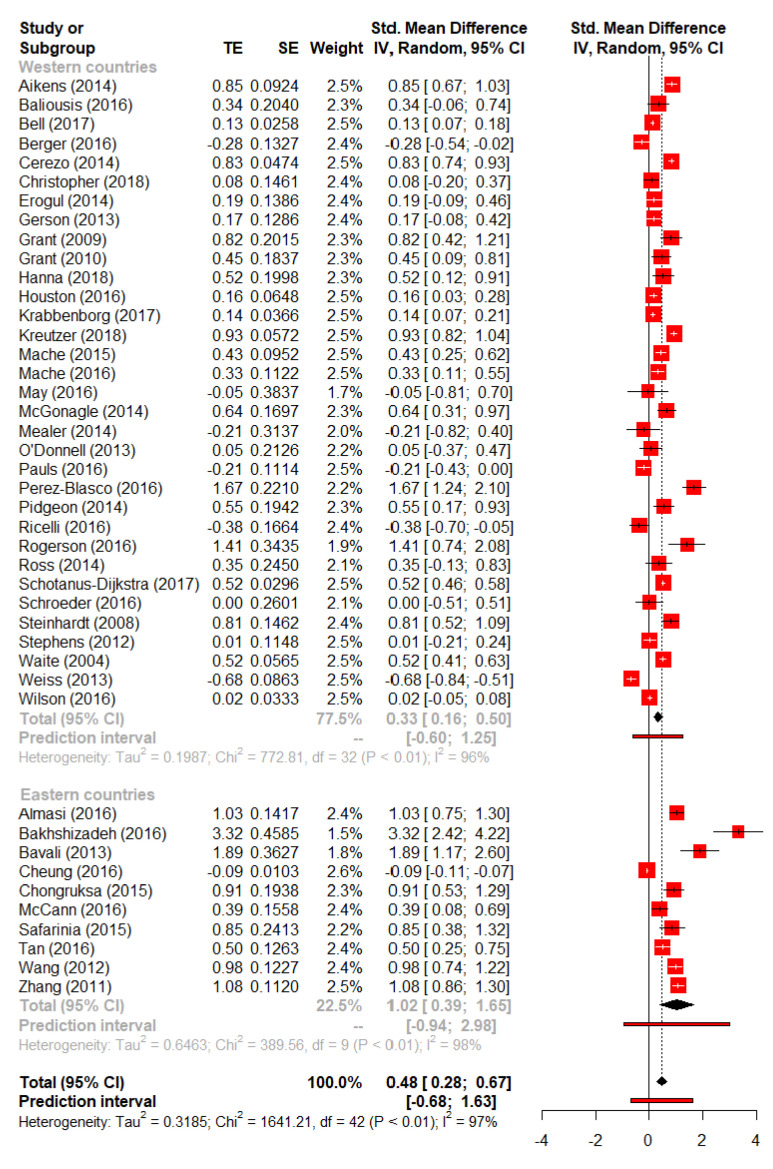
Forest plot for resilience.

**Table 1 ijerph-19-06913-t001:** Overview of the countries the studies were conducted in.

Country	Western/Eastern	*k =* 221 (100%)
USA [39,40,41,42,43,44,45,46,47,48,49,50,51,52,53,54,55,56,57,58,59,60,61,62,63,64,65,66,67,68,69,70,71,72,73,74,75,76,77,78,79,80,81,82,83,84,85,86,87,88,89,90,91,92,93,94,95,96,97,98,99,100,101,102,103,104,105,106,107,108,109,110,111,112,113,114,115,116,117,118,119,120,121,122,123,124,125,126,127,128,129,130,131,132,133,134,135,136,137,138,139,140]	Western	102 (46.2%)
Iran [141,142,143,144,145,146,147,148,149,150,151,152,153,154,155,156,157,158,159,160]	Eastern	20 (9.0%)
Australia [161,162,163,164,165,166,167,168,169,170,171,172,173,174,175,176,177,178,179]	Western	19 (8.6%)
China [180,181,182,183,184,185,186,187,188,189,190,191,192,193,194]	Eastern	15 (6.8%)
Canada [195,196,197,198,199,200,201,202,203]	Western	9 (4.1%)
Netherlands [204,205,206,207,208,209,210,211,212]	Western	9 (4.1%)
Germany [213,214,215,216,217,218]	Western	6 (2.7%)
United Kingdom [219,220,221,222,223,224]	Western	6 (2.7%)
Spain [225,226,227,228,229]	Western	5 (2.3%)
Israel [230,231,232]	Western	3 (1.4%)
South Korea [233,234,235]	Eastern	3 (1.4%)
Denmark [236,237]	Western	2 (0.9%)
India [238,239]	Eastern	2 (0.9%)
Italy [240,241]	Western	2 (0.9%)
Poland [242,243]	Western	2 (0.9%)
Portugal [244,245]	Western	2 (0.9%)
Sweden [246,247]	Western	2 (0.9%)
Thailand [246,247]	Eastern	2 (0.9%)
Belgium [248]	Western	1 (0.5%)
Cyprus [249]	Western	1 (0.5%)
Dominican Republic [250]	Eastern	1 (0.5%)
Finland [251]	Western	1 (0.5%)
Hungary [252]	Western	1 (0.5%)
New Zealand [253]	Western	1 (0.5%)
Sierra Leone [254]	Eastern	1 (0.5%)
Singapore [255]	Eastern	1 (0.5%)
Sri Lanka [256]	Eastern	1 (0.5%)
Taiwan [257]	Eastern	1 (0.5%)

**Table 2 ijerph-19-06913-t002:** Summary of the intervention characteristics and control group designs of the included studies.

Setting	*k* = 221 (100%)
Group	125 (56.5%)
Western [39,40,43,47,49,53,54,55,56,57,58,59,60,61,62,67,71,72,73,76,79,80,82,83,84,90,91,92,94,98,99,100,101,102,103,106,109,111,113,115,116,117,119,122,123,124,128,130,132,133,135,136,137,138,140,162,166,168,170,171,172,173,175,177,178,196,197,198,199,201,203,207,210,211,213,214,215,221,224,226,229,230,231,245,249,251,261,262]	88 (39.8%)
Eastern [141,142,144,145,146,147,149,150,151,152,153,154,155,156,157,158,160,181,182,183,184,185,186,187,188,190,191,193,194,234,235,246,250,254,256,257]	37 (16.7%)
Individual	44 (19.9%)
Western [42,44,45,46,48,51,52,64,69,70,77,89,93,97,107,114,118,120,121,125,126,161,163,167,195,206,212,216,218,220,222,225,232,236,240,241,242,243,244,248,253]	41 (18.6%)
Eastern [233,238,249]	3 (1.4%)
Combination	43 (19.5%)
Western [41,63,66,68,74,75,78,81,85,86,87,88,95,96,104,105,108,110,112,127,129,131,134,139,164,165,174,179,200,202,204,208,209,217,219,223,227,228,237,252]	40 (18.1%)
Eastern [180,192,239]	3 (1.4%)
Unspecified	9 (4.1%)
Western [50,65,169,176,205]	5 (2.3%)
Eastern [143,148,159,257]	4 (1.8%)
**Mode of Delivery**	***k* = 221 (100%)**
Face to face	146 (66.1%)
Western [39,40,43,47,48,49,53,54,55,56,57,58,59,60,62,64,66,67,71,72,73,74,76,79,81,82,83,84,85,90,92,94,95,98,99,100,101,102,103,104,106,109,111,113,114,115,116,117,118,119,122,123,126,128,130,131,132,133,135,136,137,138,139,140,162,164,165,166,168,170,171,175,178,196,197,198,199,201,202,204,205,207,208,210,211,212,213,214,215,217,219,220,221,223,224,226,229,230,231,240,244,245,249,251,252,253,261,262]	108 (48.9%)
Eastern [141,142,144,145,146,147,148,149,150,151,152,153,154,155,156,157,158,160,180,181,182,183,185,186,188,189,190,191,193,194,234,239,246,250,254,255,256,257]	38 (17.2%)
Combination	45 (20.4%)
Western [41,42,45,46,61,63,68,75,77,78,80,86,87,88,91,93,96,105,108,110,120,121,124,125,127,167,173,174,176,177,195,200,209,222,227,228,236,237]	38 (17.2%)
Eastern [184,187,192,233,235,238,249]	7 (3.2%)
Online	15 (6.8%)
Western [51,69,89,97,129,134,161,172,179,203,216,218,225,242,243]	15 (6.8%)
Telephone	5 (2.3%)
Western [52,70,107,112,232]	5 (2.3%)
Laboratory	3 (1.4%)
Western [44,50,250]	3 (1.4%)
Unspecified	3 (1.4%)
Western [169]	1 (0.5%)
Eastern [143,159]	2 (0.9%)
Smartphone	2 (0.9%)
Western [163,241]	2 (0.9%)
Bibliotherapy	1 (0.5%)
Western [206]	1 (0.5%)
Audio	1 (0.5%)
Western [65]	1 (0.5%)
**Population**	***k* = 221 (100%)**
Employees in organizations of different branches	51 (23.1%)
Western [41,53,55,61,62,70,72,78,80,85,91,92,97,101,103,107,108,110,117,126,127,132,135,137,138,161,164,166,168,169,170,171,175,202,208,209,213,214,215,216,222,224,227,230,237,241,242,243,251]	49 (22.2%)
Eastern [182,189]	2(0.9%)
Patients (physical health conditions)	44 (19.9%)
Western [42,45,46,47,48,54,68,74,93,96,104,113,116,118,125,133,134,136,140,167,172,195,196,197,203,210,221,226,240,244,245,252,253]	33 (14.9%)
Eastern [155,181,183,184,187,188,192,233,234,235,257]	11 (5%)
Students	32 (14.5%)
Western [43,44,49,50,51,60,65,66,67,73,76,84,89,98,99,109,111,115,120,124,128,129,131,198,200,201,207,219]	28 (12.7%)
Eastern [141,157,160,238]	4 (1.8%)
Military/police	26 (11.8%)
Western [39,40,52,57,58,59,63,64,71,82,83,90,106,119,121,122,139,162,173,174,249,261,262]	23 (10.4%)
Eastern [180,190,248]	3 (1.4%)
General population (e.g., volunteers)	25 (11.3%)
Western [69,112,114,123,176,178,205,206,212,223,225,228,232,236,250]	15 (6.8%)
Eastern [145,149,186,193,194,239,250,254,255,256]	10 (4.5%)
Family caregivers	21 (9.5%)
Western [56,75,79,94,163,179,229]	7 (3.2%)
Eastern [142,144,146,147,148,150,151,152,153,154,156,158,191,247]	14 (6.3%)
Patients (mental health conditions)	12 (5.4%)
Western [81,88,100,102,185,204,211,217,218,220]	9 (4.1%)
Eastern [143,159]	3 (1.4%)
Employees—teachers	10 (4.5%)
Western [77,86,87,95,105,130,165,177,199,231]	10 (4.5%)
**Theoretical Approach of the Intervention**	***k* = 221 (100%)**
Multimodal resilience training (several theoretical approaches); several resilience factors trained without naming certain theoretical approaches	131 (59.3%)
Western [39,40,43,44,45,46,47,49,51,52,53,54,55,57,58,59,60,64,66,70,71,72,73,74,75,76,78,79,80,85,88,89,90,95,97,98,99,100,101,102,103,106,107,108,110,111,112,113,116,117,118,123,128,129,130,131,132,134,135,136,137,162,164,165,166,168,169,170,171,173,174,175,197,200,201,202,205,207,208,210,212,213,214,215,217,220,222,223,224,226,229,230,231,236,240,245,248,249,251,252,253]	101 (45.7%)
Eastern [141,146,148,151,152,153,155,156,157,158,159,180,181,182,183,184,188,189,190,191,193,194,239,246,250,254,255,256,257]	30 (13.6%)
Mindfulness-based therapy	30 (13.6%)
Western [41,56,62,63,65,67,68,69,81,86,87,91,92,105,114,115,133,139,140,196,199,203,204,209,216,219,228,237]	28 (12.7%)
Eastern [192,238]	2 (0.9%)
CBT	23 (10.4%)
Western [93,94,120,121,124,161,163,172,176,177,179,198,211,218,221,225,242,243,244]	19 (8.6%)
Eastern [142,147,154,249]	4 (1.8%)
Unspecified	15 (6.8%)
Western [48,109,122,138,195,227,247]	7 (3.2%)
Eastern [143,144,145,149,150,186,233,235]	8 (3.6%)
AIT	6 (2.7%)
Western [61,96,125,126,127]	5 (2.3%)
Eastern [187]	1 (0.5%)
Stress inoculation training	6 (2.7%)
Western [82,83,178,232,241,246]	6 (2.7%)
Problem-solving training	3 (1.4%)
Western [84,119]	2 (0.9%)
Eastern [160]	1 (0.5%)
ACT	3 (1.4%)
Western [77,104,167]	3 (1.4%)
Positive Psychology	3 (1.4%)
Western [42,206]	2 (0.9%)
Eastern [185]	1 (0.5%)
Cognitive bias modification	1 (0.5%)
Western [50]	1 (0.5%)
**Duration**	***k* = 221 (%)**
High intensity	86 (38.9%)
Western [39,54,62,63,67,68,73,74,77,81,87,90,95,100,102,103,105,108,111,116,117,124,130,133,135,136,137,139,140,166,168,169,173,174,177,196,198,201,202,203,204,208,209,210,213,214,215,219,223,224,228,229,230,231,236,244,251,252,261,262]	60 (27.1%)
Eastern [141,144,145,147,148,151,155,157,158,160,180,183,184,185,186,188,190,192,194,234,246,247,254,255,256,257]	26 (11.8%)
Moderate intensity	80 (36.2%)
Western [41,42,43,45,46,52,55,56,57,58,60,64,66,70,72,75,78,79,88,91,92,93,94,96,101,104,106,107,109,110,112,114,115,118,119,123,128,132,134,138,161,163,164,165,167,170,172,179,197,199,200,206,211,216,218,225,226,237,240,242,245]	61 (27.6%)
Eastern [142,143,146,149,150,152,153,154,156,159,181,182,187,189,191,193,233,239,250]	19 (8.6%)
Low intensity	47 (21.3%)
Western [40,44,47,48,49,50,51,53,59,61,65,69,71,76,80,82,83,84,85,89,97,98,99,113,120,121,125,126,127,129,131,171,175,178,195,207,212,217,220,221,222,232,241,248,249,253]	46 (20.8%)
Eastern [238]	1 (0.5%)
Unspecified	8 (3.6%)
Western [86,122,162,176,205,227,243]	7 (3.2%)
Eastern [235]	1 (0.5%)
**Control Group Design**	***k* = 221 (%)**
Waitlist Control	52 (23.5%)
Western [41,42,49,51,53,62,66,68,73,76,78,85,86,87,90,91,92,93,95,96,105,106,107,109,114,115,116,126,127,128,138,161,164,172,179,195,199,204,218,226,227,228,229,230,237]	46 (20.8%)
Eastern [148,154,157,182,235,252]	6 (2.7%)
No Intervention	52 (23.5%)
Western [55,59,60,63,67,69,70,79,80,84,94,101,108,110,117,124,130,131,132,135,137,166,168,171,198,200,202,213,214,215,225,244]	32 (14.5%)
Eastern [141,142,143,144,145,146,147,149,150,151,152,153,155,159,160,180,181,186,190,191]	20 (9%)
TAU	43 (19.5%)
Western [39,47,52,54,58,64,72,75,88,100,112,113,123,139,167,173,174,175,177,197,201,205,206,210,219,223,236,240,245,249,253,261,262]	33 (14.9%)
Eastern [187,189,192,233,234,238,239,256,257,259]	10 (4.5%)
Attention Control	34 (15.4%)
Western [40,44,48,50,56,65,81,89,97,98,99,102,103,104,118,122,129,134,136,140,176,178,203,207,211,216,217,222,231,232,241,243,248]	33 (14.9%)
Eastern [193]	1 (0.5%)
Active Control	29 (13.1%)
Western [43,45,46,57,61,74,82,83,111,120,121,125,133,162,163,170,196,208,209,212,220,224,253]	23 (19.4%)
Eastern [183,184,188,194,248,258]	6 (2.7%)
unspecified	9 (4.1%)
Western [71,77,169,221,242,254]	6 (2.7%)
Eastern [156,158,185]	3 (1.4%)
Combination	2 (0.9%)
Western [119]	1 (0.5%)
Eastern [247]	1 (0.5%)

**Table 3 ijerph-19-06913-t003:** Summary of pooled effect sizes.

Outcome	*SMD*	95% *CI*	*k* (Studies)	*N* (Participants)	*I* ^2^
Resilience Western	0.33	0.16 to 0.50	33	3346	95.9%
Resilience Eastern	1.02	0.39 to 1.65	10	1451	97.7%
Anxiety Western	−0.32	−0.55 to −0.08	24	2157	94.5%
Anxiety Eastern	−0.67	−0.76 to −0.58	5	279	0%
Depressive symptoms Western	−0.44	−0.62 to −0.26	14	1576	91.1%
Depressive symptoms Eastern	−0.55	−2.96 to 1.86	2	130	66.4%
Quality of life Western	0.28	−0.13 to 0.44	33	4610	98.3%
Quality of life Eastern	0.37	−0.03 to 0.76	5	1135	85.1%
Perceived stress Western	−0.41	−0.61 to −0.20	35	2789	91.4%
Perceived stress Eastern	−0.71	−3.74 to 2.32	2	129	85.0%
Social support Western	−0.12	−2.56 to 2.32	2	275	89.4%
Social support Eastern	0.21	−2.67 to 3.09	2	1314	99.7%

## Data Availability

All relevant data generated or analyzed during this review are included in this published article. More detailed extracted data from the included studies are available upon request from the corresponding author.

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
