# Peer review of "Resilience Interventions Conducted in Western and Eastern Countries—A Systematic Review"

_ijerph, 2022, doi:10.3390/ijerph19116913_

Round 1
Reviewer 1 Report
This review compares the efficacy of resilience intervention on resilience and other psychological outcomes between Western and Eastern countries. The authors compare intervention delivery, target population, study design, and theoretical approach. While this review addresses the gap about geographical consideration, there a number of issues warrants concerns:
First, it seems that the authors overgeneralize the differences in cultural context by dichotomizing into Western and Eastern countries. I would suggest being more cautious in claiming geographical differences (by country), rather than cultural context differences without operationalizing any relevant variables. For example, even in the U.S., Americans of different race/ethnicity embrace different cultural values.
Second, the focus on mental health is a fundamental western concept. The awareness of own mental health symptoms requires individuals’ insights. It would create biases when it comes to self-report of mental health outcomes. In addition, I encourage the authors to look at the literature about somatization of mental health symptoms, which is common among people from Eastern countries. It is biased without considering the obvious differences in the manifestation of mental health symptoms across Western and Eastern countries.
It is disappointing that the authors do not explicitly define resilience as a process or outcome for the scope of this review study and the study screening process. As the understanding of resilience is controversial in the field, to some, resilience is considered as a capacity triggered by stressors. The authors are suggested to clearly define their conceptualization of resilience of adults. Moreover, it is significant to systematically review how each resilience intervention defines and operationalizes resilience; otherwise, it is very difficult for the authors to compare the resilience interventions.
For the methods, the authors follow a rigorous process of literature search and screening process. However, it seems that the search was performed in June 2019. Since it has been almost 3 years, which is outdated, the authors are encouraged to do an updated search and include recently published studies.
Since the authors include both prevention (target people who are anticipated to be exposed to adversities in the future) and intervention (being exposed to stressors in the past or currently) programs, this will impact the perceived stress level, resilience, and mental health outcomes. It is difficult to trigger the resilience capacity without being exposed to stressors. Thus, it is important for the authors to take into account the period being exposed to stressors and the nature of stressors (chronic vs acute, cumulative vs isolated, normative vs unexpected).
Boss, P., Bryant, C. M., & Mancini, J. A. (2016). Family stress management: A contextual approach. Sage Publications.
Since the authors broadly include intervention/prevention programs targeting adults facing past/current/future stressors of any nature, it is unreasonable for the authors to exclude resilience intervention targeting adults facing on-going pandemic stressors.
For the results, it seems that the authors are comparing the effect sizes of different psychological outcomes across different interventions. This is similar to a meta-analysis. However, the authors did not take into account of the development stage of intervention/prevention programs and sample size of each intervention. It is not reasonable to compare a pilot RCT study with a small sample size. Thus, it is important for the authors to restrict the development stage of intervention/prevention programs and set a minimum sample size.
Moreover, different resilience intervention/prevention programs may target resilience as a primary outcome, and other psychological well-being as secondary outcomes. It is inappropriate to compare the primary outcomes of one intervention with the secondary outcomes of another intervention, even looking at the same psychological well-being construct. Thus, it is important to evaluate the intervention/prevention outcomes based on the purpose/aim of the intervention/prevention programs.
The discussion is too brief. The authors need to demonstrate (a) how this review contributes to the existing systematic reviews of resilience intervention, and (b) implications to future research and practitioners in different cultural contexts.
Existing review of resilience interventions:
Liu, J. J. W., Ein, N., Gervasio, J., Battaion, M., & Fung, K. (2022). The Pursuit of Resilience: A Meta-Analysis and Systematic Review of Resilience-Promoting Interventions. Journal of Happiness Studies, 23, 1771-1791
Ferreira, M., Marques, A., & Gomes, P. V. (2021). Individual Resilience Interventions: A Systematic Review in Adult Population Samples over the Last Decade. International journal of environmental research and public health, 18(14), 7564.
Díaz-García, A., Franke, M., Herrero, R., Ebert, D. D., & Botella, C. (2021). Theoretical adequacy, methodological quality and efficacy of online interventions targeting resilience: a systematic review and meta-analysis. European journal of public health, 31(Supplement_1), i11-i18.
Chmitorz, A., Kunzler, A., Helmreich, I., Tüscher, O., Kalisch, R., Kubiak, T., ... & Lieb, K. (2018). Intervention studies to foster resilience–A systematic review and proposal for a resilience framework in future intervention studies. Clinical psychology review, 59, 78-100.
Joyce, S., Shand, F., Tighe, J., Laurent, S. J., Bryant, R. A., & Harvey, S. B. (2018). Road to resilience: a systematic review and meta-analysis of resilience training programmes and interventions. BMJ open, 8(6), e017858.
Author Response
We want to thank the Reviewer for their helpful and detailed notes to our paper and the suggestions that were made to improve it. We commented each note below and updated our paper accordingly.
- First, it seems that the authors overgeneralize the differences in cultural context by dichotomizing into Western and Eastern countries. I would suggest being more cautious in claiming geographical differences (by country), rather than cultural context differences without operationalizing any relevant variables. For example, even in the U.S., Americans of different race/ethnicity embrace different cultural values.
- We agree and now addressed this point in our discussion under limitations (starting line 371) and at Implications (starting line 415)
- Second, the focus on mental health is a fundamental western concept. The awareness of own mental health symptoms requires individuals’ insights. It would create biases when it comes to self-report of mental health outcomes. In addition, I encourage the authors to look at the literature about somatization of mental health symptoms, which is common among people from Eastern countries. It is biased without considering the obvious differences in the manifestation of mental health symptoms across Western and Eastern countries.
- Thank you and we added this starting at line 66 in the introduction as well as in the discussion starting at line 411.
- It is disappointing that the authors do not explicitly define resilience as a process or outcome for the scope of this review study and the study screening process. As the understanding of resilience is controversial in the field, to some, resilience is considered as a capacity triggered by stressors. The authors are suggested to clearly define their conceptualization of resilience of adults. Moreover, it is significant to systematically review how each resilience intervention defines and operationalizes resilience; otherwise, it is very difficult for the authors to compare the resilience interventions.
- Thank you and we added a more detailed resilience definition starting at Line 31.
“Resilience is a lifelong, ongoing process that does not necessarily lead to a person encountering fewer stressors in their life, but can lead to more effective coping of stressors and more adaptive responses [3]. The most recent approach in social and health science conceptualizes resilience as a positive outcome, i.e. maintaining or quick regaining mental health during or after adversities (1). In this concept, adaptation processes that finally lead to resilience outcomes are called resilience processes, and those resilience processes are facilitated by resilience factors. Resilience factors may be psychosocial factors (e.g. active coping, self-efficacy, optimism, social support and hope), but also (epi)genetic, neurobiological, immunological or other biological factors [6], which are associated to each other and may interact.“
- For the second part of this comment (“Moreover, it is significant to systematically review how each resilience intervention defines and operationalizes resilience; otherwise, it is very difficult for the authors to compare the resilience interventions”) we added the lines starting at 226.
“Of all 221 included studies, 43 studies measured resilience as an outcome. In these studies, resilience was defined as a state or process most of the times (N = 41, assessed for example with CD-RISC or BRS), whereas only two studies defined resilience as a trait (assessed with the Cognitive Hardiness Scale by Nowack 1989 [263]).”
- For the methods, the authors follow a rigorous process of literature search and screening process. However, it seems that the search was performed in June 2019. Since it has been almost 3 years, which is outdated, the authors are encouraged to do an updated search and include recently published studies.
- It is true that our very comprehensive literature search was performed in 2019. However, since the primary goal of the review was to compare interventions in western and eastern countries and since we identified a relatively large number of studies to successfully compare the efficacy in both groups of countries, we believe that the study can rely on this literature identified until 2019.
- Since the authors include both prevention (target people who are anticipated to be exposed to adversities in the future) and intervention (being exposed to stressors in the past or currently) programs, this will impact the perceived stress level, resilience, and mental health outcomes. It is difficult to trigger the resilience capacity without being exposed to stressors. Thus, it is important for the authors to take into account the period being exposed to stressors and the nature of stressors (chronic vs acute, cumulative vs isolated, normative vs unexpected).
Boss, P., Bryant, C. M., & Mancini, J. A. (2016). Family stress management: A contextual approach. Sage Publications.
- Thank you and we added a narrative description of the targeted population and the linked stressors starting at line 229:
“In these studies, the intervention took mostly place during a stressor (N = 39), and partly before an anticipated stressor (N = 1), after a stressor (N = 2), or unspecified (N = 2). Some stressors were normative (N = 13, e.g. workplace related stress, age-associated loss of resources or academic stress), and some were non-normative (N = 30, e.g. sudden severe illnesses, nature disasters or homelessness).”
- We also added a sentence at the discussion part (starting at line 381) since we couldn’t conclude further analysis regarding this aspect due to the small number of subgroups in each outcome.
- Since the authors broadly include intervention/prevention programs targeting adults facing past/current/future stressors of any nature, it is unreasonable for the authors to exclude resilience intervention targeting adults facing on-going pandemic stressors.
- As outlined and as we know from our ongoing research, resilience interventions during the COVID-19-pandemic are mostly still not published, which let us not to include parts of them before getting a comprehensive picture and also a sufficient number to compare interventions between western and eastern countries.
- For the results, it seems that the authors are comparing the effect sizes of different psychological outcomes across different interventions. This is similar to a meta-analysis. However, the authors did not take into account of the development stage of intervention/prevention programs and sample size of each intervention. It is not reasonable to compare a pilot RCT study with a small sample size. Thus, it is important for the authors to restrict the development stage of intervention/prevention programs and set a minimum sample size.
- Indeed, we performed also meta-analyses if possible. The sample size of an intervention is in fact relevant for the effect size estimation and the confidence intervals of the estimations. However, since the sample size directly influences the confidence intervals, there is no need to set a minimum sample size. Sample sizes of the intervention and control groups are given in Table S4.
- We also added a sentence to address the sample size of the RCTs included in the analysis for the outcome resilience (starting line 234)
“The sample size of these 43 studies ranged from n = 22 to n = 918 with a mean sample size of n = 110.63 (Table S4).”
- Moreover, different resilience intervention/prevention programs may target resilience as a primary outcome, and other psychological well-being as secondary outcomes. It is inappropriate to compare the primary outcomes of one intervention with the secondary outcomes of another intervention, even looking at the same psychological well-being construct. Thus, it is important to evaluate the intervention/prevention outcomes based on the purpose/aim of the intervention/prevention programs.
- We added the following sentences (starting at line 388) that hopefully answers this comment:
“As suggested by Copas et al. [273], the reviewed studies include primary as well as secondary outcomes. We didn‘t take into account in our analysis if the outcomes were originally assessed as primary or as secondary outcomes. Future research might com-pare if the efficacy of resilience and other outcomes differ when assessed as primary or as secondary outcomes in reviewed studies.”
- The discussion is too brief. The authors need to demonstrate (a) how this review contributes to the existing systematic reviews of resilience intervention, and (b) implications to future research and practitioners in different cultural contexts.
- We rewrote the discussion it now has the following parts included: principal findings, comparison to literature, limitations, implications, conclusion. We hope these additions have led the discussion in a better direction.
- We included the reviews that you suggested from Liu et al., Joyce et al., and mentioned Chmitorz et al. in the “Comparison to Literature” part and want to thank you for the input!
Existing review of resilience interventions:
Liu, J. J. W., Ein, N., Gervasio, J., Battaion, M., & Fung, K. (2022). The Pursuit of Resilience: A Meta-Analysis and Systematic Review of Resilience-Promoting Interventions. Journal of Happiness Studies, 23, 1771-1791
Ferreira, M., Marques, A., & Gomes, P. V. (2021). Individual Resilience Interventions: A Systematic Review in Adult Population Samples over the Last Decade. International journal of environmental research and public health, 18(14), 7564.
Díaz-García, A., Franke, M., Herrero, R., Ebert, D. D., & Botella, C. (2021). Theoretical adequacy, methodological quality and efficacy of online interventions targeting resilience: a systematic review and meta-analysis. European journal of public health, 31(Supplement_1), i11-i18.
Chmitorz, A., Kunzler, A., Helmreich, I., Tüscher, O., Kalisch, R., Kubiak, T., ... & Lieb, K. (2018). Intervention studies to foster resilience–A systematic review and proposal for a resilience framework in future intervention studies. Clinical psychology review, 59, 78-100.
Joyce, S., Shand, F., Tighe, J., Laurent, S. J., Bryant, R. A., & Harvey, S. B. (2018). Road to resilience: a systematic review and meta-analysis of resilience training programmes and interventions. BMJ open, 8(6), e017858.
Reviewer 2 Report
I appreciate the possibility to review this very interesting paper. It deals with an important issue of resilience in culture related context. The work put into the preparation of the article is very profound and the presentation is clear. There are, however, some suggestions and notes I would like to point out:
- In Introduction there is and important problem - resilience must be understood as a concept not only related to trauma, but also as a personality trait and as a process. Please, clarify the definition you used while conducting your research.
- It would be appreciated if the Authors explain in more detailed way what would be the possible impact of the knowledge gained by their study.
- The description of purpose of the research should be replaced according to standards, it rather belongs to the section material and method.
- The very important issue is not taking onto consideration the type of crisis (normative or non-normative) and the modality of it. Please, ad this to your text.
- Please, explain what is the source of your classification of interventions. It is not separable and it is not clear on what basis it was created.
- My another serious concern is the discussion of the results obtained in the study. This part is rather a repetition of the research results than their discussion in the context of the intended goals and results.
- Please, explain the ambiguity concerning cultural context. What was your main goal- to compare cultures? not to do it? It is not clear in your paper. Please, address this issue.
Thank you for the opportunity to review the paper.
Author Response
We want to thank the Reviewer for their kind words and helpful notes to our paper. We commented each note below and updated our paper accordingly to your suggestions.
I appreciate the possibility to review this very interesting paper. It deals with an important issue of resilience in culture related context. The work put into the preparation of the article is very profound and the presentation is clear. There are, however, some suggestions and notes I would like to point out:
1. In Introduction there is and important problem - resilience must be understood as a concept not only related to trauma, but also as a personality trait and as a process. Please, clarify the definition you used while conducting your research.
- We added a more detailed resilience definition starting at Line 31.
“Resilience is a lifelong, ongoing process that does not necessarily lead to a person encountering fewer stressors in their life, but can lead to more effective coping of stressors and more adaptive responses [3]. The most recent approach in social and health science conceptualizes resilience as a positive outcome, i.e. maintaining or quick regaining mental health during or after adversities (1). In this concept, adaptation processes that finally lead to resilience outcomes are called resilience processes, and those resilience processes are facilitated by resilience factors. Resilience factors may be psychosocial factors (e.g. active coping, self-efficacy, optimism, social support and hope), but also (epi)genetic, neurobiological, immunological or other biological factors [6], which are associated to each other and may interact.“
2. It would be appreciated if the Authors explain in more detailed way what would be the possible impact of the knowledge gained by their study.
- We elaborated on the possible impact of the paper further starting at line 85 and added a “implications” part in the discussion.
3. The description of purpose of the research should be replaced according to standards, it rather belongs to the section material and method.
- We changed the purpose of research-part and moved some of the text-parts that would better fit to the material and methods-part, starting line 89:
“In this review we compared different aspects of resilience interventions between training programs conducted in Western versus Eastern countries, specifically study settings, mode of delivery, target populations, underlying theoretical approaches, du-rations of training, as well as the study design by regarding the control group designs that were used. We also contrasted their efficacy for a range of mental health outcomes. More specifically, the effects of resilience-training programs conducted in Western and Eastern countries on the outcomes of resilience, anxiety, depressive symptoms, quality of life, perceived stress and social support were examined in this review. We assessed both resilience and resilience related outcomes as resilience is no unified construct in the current and past research field and we don’t want to exclude studies just because they had a different wording for a similar construct in different countries and time periods.”
4. The very important issue is not taking onto consideration the type of crisis (normative or non-normative) and the modality of it. Please, ad this to your text.
- We added a narrative part for the type of crisis the different target population groups were going through, starting at line 229.
“In these studies, the intervention took mostly place during a stressor (N = 39), and partly before an anticipated stressor (N = 1), after a stressor (N = 2), or unspecified (N = 2). Some stressors were normative (N = 13, e.g. workplace related stress, age-associated loss of resources or academic stress), and some were non-normative (N = 30, e.g. sud-den severe illnesses, nature disasters or homelessness).”
5. Please, explain what is the source of your classification of interventions. It is not separable and it is not clear on what basis it was created.
- We included studies that explicitly state that the aim was to foster resilience, hardiness or post‐traumatic growth (starting at line 127)
6. My another serious concern is the discussion of the results obtained in the study. This part is rather a repetition of the research results than their discussion in the context of the intended goals and results.
- We rewrote the discussion so that it now includes the following parts: principal findings, comparison to literature, limitations, implications, conclusion. We hope these additions have led the discussion in a better direction.
7. Please, explain the ambiguity concerning cultural context. What was your main goal- to compare cultures? not to do it? It is not clear in your paper. Please, address this issue.
- We hope our additions and rewriting under the “purpose of review” section and also in the discussion section are now bringing out our goals more clearly.
Thank you for the opportunity to review the paper.
Reviewer 3 Report
I appreciate reading your article. This article has the potential to give us a much better understanding of implemented interventions in the area of resilience between the countries of the East and the West. I recognise the contribution of the work you have put into the analyzes you have carried out and the preparation of the summaries that you have presented in a clear form in tables and figures. However, I think that before this article can be published, several important factors should be taken into account, which I mention below:
- Authors state (lines 64-66) that “Therefore, this review aimed to address this research gap and to draw attention to the WEIRD problem by contrasting resilience interventions for adult populations in Western and Eastern countries.” - it would be valuable to expand your rationale - it would show the comparison of the interventions, emphasize why it is so important in the case you are analyzing, which will let us know and what this knowledge can contribute to.
- Lines 66-73 describe what authors have analyzed. I would recommend rewriting this part to better reflect the purpose of your research, or transferring these lines to the materials and method part.
- Regarding Introduction and lines 106-108, authors state that “We included psychological resilience interventions, that is, interventions focusing on fostering resilience or related concepts (e.g., hardiness or posttraumatic growth) [22,23]”. Resilience is a broader concept than posttraumatic growth and is different from hardiness. Although it happens that in literature it is equated with hardiness and treated as the same concept. Resilience is also understood as a personality trait and as a process. It is not clear how resilience defined by authors? It will be valuable to explain why authors did not focus only on the resilience aspect and included similar concepts in its scope - I would recommend extending the Introduction to this aspect, so that the reader will gain a broader perspective. This will help clarify the definition of resilience used in the article.
- The characterization of the stressors that people struggle with is a fundamental aspect when discussing resilience. The authors summarized interventions with regards to: setting, mode of delivery, population, theoretical approach of the intervention, duration, control group design, but they did not take into account the types of crises and problems in which interventions were undertaken, which is a major shortcoming in the article.
- Its not clear wheather the authors have taken into account the direction of the scales when calculating (SMD) the same area, e.g. resilience or anxiety? Did the high scores on the various questionnaires always indicate a high intensity of a given feature?
- Authors state “Training length of interventions could be clustered into high intensity (i.e., more than 12 hours or 12 sessions), moderate intensity (i.e., 5 to 12 hours or 3 to 12 sessions) and low intensity (i.e., less than 5 hours total or 3 sessions)”. What was the basis this classification made?
- The discussion needs to be rewritten. The authors replicate their results in the section above. The discussion requires explaining why it was researched, what results were obtained and why it is needed and important. It is also necessary to refer to the literature on the topic.
- Authors state that: “The intervention studies included in this review did not focus on cultural aspects and therefore did not specifically describe this aspect nor were the subjects asked for a self-assessment about the country they live – or the culture they hold - in. Therefore, we had to work with the country label of each study and could not describe this aspect of the target populations further”. I would recommend the authors delete this passage. Research in a given country did not have to focus on the cultural aspect. Research that is carried out in different countries, in principle, examines various features revealed in a given culture.
- I would recommend renaming the Outlook part to Conclusions, expanding it and listing it in sub-items.
Author Response
We want to thank the Reviewer for their helpful comments to our paper. We commented each note below and updated our paper accordingly.
I appreciate reading your article. This article has the potential to give us a much better understanding of implemented interventions in the area of resilience between the countries of the East and the West. I recognise the contribution of the work you have put into the analyzes you have carried out and the preparation of the summaries that you have presented in a clear form in tables and figures. However, I think that before this article can be published, several important factors should be taken into account, which I mention below:
1. Authors state (lines 64-66) that “Therefore, this review aimed to address this research gap and to draw attention to the WEIRD problem by contrasting resilience interventions for adult populations in Western and Eastern countries.” - it would be valuable to expand your rationale - it would show the comparison of the interventions, emphasize why it is so important in the case you are analyzing, which will let us know and what this knowledge can contribute to.
- Thank you and we added this starting at line 66 and also starting at line 85
2. Lines 66-73 describe what authors have analyzed. I would recommend rewriting this part to better reflect the purpose of your research, or transferring these lines to the materials and method part.
- Thank you for your thoughtful reading, we transferred these lines to the materials and method-part (starting line 88).
3. Regarding Introduction and lines 106-108, authors state that “We included psychological resilience interventions, that is, interventions focusing on fostering resilience or related concepts (e.g., hardiness or posttraumatic growth) [22,23]”. Resilience is a broader concept than posttraumatic growth and is different from hardiness. Although it happens that in literature it is equated with hardiness and treated as the same concept. Resilience is also understood as a personality trait and as a process. It is not clear how resilience defined by authors? It will be valuable to explain why authors did not focus only on the resilience aspect and included similar concepts in its scope - I would recommend extending the Introduction to this aspect, so that the reader will gain a broader perspective. This will help clarify the definition of resilience used in the article.
- We elaborated on the resilience definition further, starting at Line 31:
“Resilience is a lifelong, ongoing process that does not necessarily lead to a person encountering fewer stressors in their life, but can lead to more effective coping of stressors and more adaptive responses [3]. The most recent approach in social and health science conceptualizes resilience as a positive outcome, i.e. maintaining or quick regaining mental health during or after adversities (1). In this concept, adaptation processes that finally lead to resilience outcomes are called resilience processes, and those resilience processes are facilitated by resilience factors. Resilience factors may be psychosocial factors (e.g. active coping, self-efficacy, optimism, social support and hope), but also (epi)genetic, neurobiological, immunological or other biological factors [6], which are associated to each other and may interact.“
4. The characterization of the stressors that people struggle with is a fundamental aspect when discussing resilience. The authors summarized interventions with regards to: setting, mode of delivery, population, theoretical approach of the intervention, duration, control group design, but they did not take into account the types of crises and problems in which interventions were undertaken, which is a major shortcoming in the article.
- We added a narrative part for the type of crisis the different target population groups were going through, starting at line 229.
5. Its not clear wheather the authors have taken into account the direction of the scales when calculating (SMD) the same area, e.g. resilience or anxiety? Did the high scores on the various questionnaires always indicate a high intensity of a given feature?
- Thank you and yes, the higher SMDs in our paper indicate a higher intensity of the named outcome. Thus, higher SMDs in the outcome resilience can be considered positive, while lower SMDs in the outcome anxiety would indicate that this outcome has decreased in the target group.
- We added a sentence starting at line 170 to clarify this aspect in the paper.
6. Authors state “Training length of interventions could be clustered into high intensity (i.e., more than 12 hours or 12 sessions), moderate intensity (i.e., 5 to 12 hours or 3 to 12 sessions) and low intensity (i.e., less than 5 hours total or 3 sessions)”. What was the basis this classification made?
- Many thanks for this comment. This subgroup analysis regarding training length was also performed in our Cochrane review on psychological resilience interventions in healthcare staff (Kunzler et al., 2020). Based on the evidence synthesized here as well as the primary studies included in previous systematic reviews (e.g., Leppin et al., 2014), we chose this classification since 8-hour/8-session training programs were implemented most frequently. To ensure comparability with the Cochrane review, this classification of training length was also used in the current review focusing on intercultural differences.
7. The discussion needs to be rewritten. The authors replicate their results in the section above. The discussion requires explaining why it was researched, what results were obtained and why it is needed and important. It is also necessary to refer to the literature on the topic.
- We rewrote the discussion and it now includes the following parts: principal findings, comparison to literature, limitations, implications, conclusion. We hope these additions have led the discussion in a better direction.
8. Authors state that: “The intervention studies included in this review did not focus on cultural aspects and therefore did not specifically describe this aspect nor were the subjects asked for a self-assessment about the country they live – or the culture they hold - in. Therefore, we had to work with the country label of each study and could not describe this aspect of the target populations further”. I would recommend the authors delete this passage. Research in a given country did not have to focus on the cultural aspect. Research that is carried out in different countries, in principle, examines various features revealed in a given culture.
- We agree with your statement and deleted the sentence as suggested.
9. I would recommend renaming the Outlook part to Conclusions, expanding it and listing it in sub-items.
- As mentioned under point 7, we rewrote the discussion-part.
Round 2
Reviewer 1 Report
Most of my earlier concerns are well-addressed. However, I encourage the authors to revise the title by removing "of intercultrual differences" as this title may be misleading about the depth of cultural comparison for this systematic review.
Author Response
We want to thank the reviewer again for their helpful comments on our paper!
- Most of my earlier concerns are well-addressed. However, I encourage the authors to revise the title by removing "of intercultrual differences" as this title may be misleading about the depth of cultural comparison for this systematic review.
As suggested by the reviewer we changed the title slightly to not mislead future readers into “Resilience interventions conducted in Western and Eastern countries – a systematic review”
We also let a native speaker proofread our paper and conducted a spell check as requested in the key points of the report form.
Reviewer 2 Report
Thank you for following the suggestions and making the improvements in your paper. However, there are some repetitions and very complicated, long sentences and therefore it is hard to understand the meaning of the several sentences (line 34-38; 52-55; 84-94). This article still demands the extensive editing of English language and style, especially the added parts of text.
Author Response
We want to thank the reviewer again for their detailed reading of our paper!
- Thank you for following the suggestions and making the improvements in your paper. However, there are some repetitions and very complicated, long sentences and therefore it is hard to understand the meaning of the several sentences (line 34-38; 52-55; 84-94). This article still demands the extensive editing of English language and style, especially the added parts of text.
As suggested, we let a native speaker proofread our paper. We also tried to simplify/split the sentences that the reviewer referred to. We hope the statements in those sections became clearer.
Reviewer 3 Report
Thank you for introducing the amendments that make the text clearer. At the same time, I would like to point out that the text requires native language proofreading.
Author Response
We want to thank the reviewer again for their helpful comments on our paper!
- Thank you for introducing the amendments that make the text clearer. At the same time, I would like to point out that the text requires native language proofreading.
As suggested, we let a native speaker proofread our paper and hope that the changes in the text improve our paper.